# Gauge Fiber Bundle Geometry of Transformers

## Abstract

We give a geometry-first account of Transformers with GeLU. On a generic regular set of parameters, the head-wise symmetry group acts freely and properly, so the parameter space fibers over a quotient of functionally distinct models—a clean principal-bundle picture with gauge orbits as fibers and function-changing directions as horizontals. Using the empirical Fisher (Fisher–Rao) metric yields a canonical horizontal distribution and clarifies that the natural gradient is the *horizontal Riesz representative* of the Euclidean gradient (reducing to orthogonal projection only in a special case). Within this framework, attention behaves like a connection with generically nonzero curvature (path-dependent transport), while the feed-forward block is largely fiber-preserving with a dimension-controlled near-orthogonality to attention. We turn these ideas into practical diagnostics—a least-squares, gauge-aware gradient split and a small-loop holonomy estimator—then report *Euclidean-proxy* consistency checks aligning with the theory; full Fisher–Rao evaluations are presented as algorithms for future work. Architectural choices such as RoPE appear as principled gauge reductions (e.g., per-head $Q/K$ dimension from $d_k^2$ to $d_k$).

## 1 Introduction

Transformer architectures exhibit extensive parameter symmetries. In particular, head-wise transformations that preserve the realized function create large gauge orbits in parameter space and obscure the geometry of optimization. A principled analysis therefore requires isolating directions tangent to gauge orbits from directions that modify the function, and studying learning on the resulting quotient manifold.

**Preliminaries.** Let $\pi : \Theta_0 \to \mathcal{Q}$ denote the parameter–function map on a regular (Zariski–open) subset $\Theta_0$ of parameters. The fibers of $\pi$ are gauge orbits of a Lie group $G$, and $T_\theta \Theta$ admits the $g_\theta$–orthogonal decomposition $T_\theta \Theta = \mathcal{V}_\theta \oplus \mathcal{H}_\theta$ into vertical directions $\mathcal{V}_\theta = \ker d\pi_\theta$ and horizontal directions $\mathcal{H}_\theta$. A principal connection specifies $\mathcal{H}_\theta$ smoothly; its connection 1–form $\omega$ has curvature $\Omega = d\omega + \frac{1}{2}[\omega, \omega]$, which governs holonomy on $\mathcal{Q}$. Throughout, the horizontal distribution is chosen as the Fisher–Rao (FR) orthogonal complement $\mathcal{H}_\theta = \mathcal{V}_\theta^{\perp_{g_\theta}}$ associated with the empirical Fisher metric $g_\theta$.

This paper provides a self–contained geometric formulation for Transformers with GeLU activations. On the regular stratum $\Theta_0$, the head–wise *maximal* symmetry group $G_{\max} = ((\mathrm{GL}(d_k))^h \times (\mathrm{GL}(d_v))^h) \rtimes S_h$ acts *freely and properly*, hence $\pi : \Theta_0 \to \mathcal{Q} := \Theta_0 / G_{\max}$ is a principal $G_{\max}$–bundle (Theorem 2.3). The FR connection yields a canonical horizontal/vertical split. In this geometry, the natural gradient is the *horizontal Riesz representative* of the Euclidean gradient,

$$\widetilde{\nabla} L \;=\; (G_{\theta|\mathcal{H}_\theta})^\dagger \, P_{\mathcal{H}_\theta}^\top \nabla L,$$

reducing to $g_\theta$–orthogonal projection only in the special case $G_{\theta|\mathcal{H}_\theta} = I$ (Theorem 3.2). The attention mechanism induces an Ehresmann connection on the representation bundle with *nonzero* curvature on a Zariski–open subset of $\Theta_0$ (Theorem 4.1), whereas the feed–forward block produces gradients that are predominantly vertical and nearly orthogonal to attention in the FR metric, with a dimension–controlled bound (Proposition 4.2).

Two diagnostic procedures follow from this framework. First, a metric–aware least–squares decomposition of a gradient into vertical and horizontal components (Algorithm 1). Second, a small–loop holonomy estimator with Richardson extrapolation whose bias is $O(\varepsilon^3)$ (Algorithm 2; Theorem 5.1). All empirical results in this submission use *Euclidean* inner products as computationally tractable proxies for FR quantities; they provide conservative validation of the predicted geometry (Section 7). FR–exact procedures are stated and analyzed but deferred to future large–scale evaluation.

The scope and assumptions are explicit. We consider standard Transformer blocks with GeLU and fixed widths. Results are proved on a generic stratum $\Theta_0$ where the relevant projection operators have full column rank; outside $\Theta_0$ stabilizers increase and the action is stratified by orbit type. Architectural variants fit naturally: rotary position embeddings (RoPE) restrict the $Q/K$ factor to the plane–wise commutant, reducing the per–head gauge dimension from $d_k^2$ to $d_k$, while grouped/multi–query attention ties symmetry factors across heads. The presentation proceeds as follows: Section 2 establishes maximal symmetry and the principal–bundle structure; Section 3 develops the FR connection and the natural gradient; Section 4 proves attention curvature and FFN near–verticality; Section 5 presents the diagnostics and their complexity; Section 6 gives the quotient Morse–Bott view; Section 7 reports Euclidean–proxy evidence. Proofs appear in the appendix.

## 2 Maximal Gauge Symmetry and the Principal Bundle

We write $h$ for the number of heads, $d_k$ and $d_v$ for key/query and value widths, and $d_{\text{model}}$ for the model dimension. A single multi-head attention (MHA) layer is parameterized by

$$\theta = \big\{ (W_Q^{(i)}, W_K^{(i)}, W_V^{(i)})_{i=1}^h, W_O \big\},$$

and a depth-$L$ model by $(\theta_1, \ldots, \theta_L)$. The parameter manifold is an open set $\Theta \subset \mathbb{R}^D$, and $\pi : \Theta \to \mathcal{Q}$ denotes the parameter–function map to the quotient of functionally distinct models.

**Definition 2.1** (Generic stratum). *Let $\Theta_0 \subset \Theta$ be the set of parameters such that, for every head $i$ in every layer:*

*(G1)* $\operatorname{rank}(W_Q^{(i)}) = \operatorname{rank}(W_K^{(i)}) = d_k$ *and* $\operatorname{rank}(W_V^{(i)}) = d_v$;

*(G2)* $d_{model} = h\, d_v$;

*(G3)* (Softmax Jacobian nondegeneracy) *for the fixed evaluation batch used to define $g_\theta$, the Jacobian of the attention weights with respect to pre-softmax logits has full row rank at every token position. Equivalently, the score map varies along at least two independent horizontal directions locally (no collapsed/constant softmax across all heads).*

*This set is Zariski–open and therefore of full measure.*

**Intuition for freeness and properness.** On $\Theta_0$, the full-column-rank conditions forbid nontrivial stabilizers: if $g = (A_i, C_i; \sigma)$ satisfies $g \cdot \theta = \theta$, then $W_Q^{(i)} A_i = W_Q^{(\sigma(i))}$, $W_K^{(i)} A_i^{-\top} = W_K^{(\sigma(i))}$, $W_V^{(i)} C_i = W_V^{(\sigma(i))}$ and $C_i^{-1} W_O = W_O$ force (by identifiability and (G3)) $A_i = C_i = I$ and $\sigma = e$, hence freeness. Properness follows from the closed-graph criterion for matrix-group actions: if $g_n \cdot \theta_n \to \theta'$ with $\theta_n \to \theta \in \Theta_0$, then the full-rank bounds imply $(g_n)$ is bounded and has a convergent subsequence; see Appendix A for details.

Two families of changes of basis preserve the realized function: invertible transforms in the $Q/K$ channels for each head, and invertible transforms in the $V$ channels paired with a compensating block in $W_O$; heads may also be permuted. The resulting symmetry group is

$$G_{\max} = \Big( (\operatorname{GL}(d_k))^h \times (\operatorname{GL}(d_v))^h \Big) \rtimes S_h,$$

acting by

$$(W_Q^{(i)}, W_K^{(i)}) \mapsto (W_Q^{(i)} A_i,\ W_K^{(i)} A_i^{-\top}), \qquad (W_V^{(i)}, W_{O,i}) \mapsto (W_V^{(i)} C_i,\ C_i^{-1} W_{O,i}),$$

for $(A_i) \in (\operatorname{GL}(d_k))^h$, $(C_i) \in (\operatorname{GL}(d_v))^h$, and by $\sigma \in S_h$ permuting heads.

**Theorem 2.2** (Maximal gauge group for a single attention layer). *On $\Theta_0$, the group of all parameter transformations that preserve the multi-head attention block equals*

$$G_{\max} = \left( (\mathrm{GL}(d_k))^h \times (\mathrm{GL}(d_v))^h \right) \rtimes S_h,$$

*acting head-wise as above. No additional continuous or discrete symmetries exist.*

*Proof reference.* Appendix A.1, which establishes sufficiency, attention-weight identifiability up to permutation, Lie-algebra characterization, block-diagonality (no cross-head mixing beyond $S_h$), and completeness.

**Theorem 2.3** (Principal bundle on the generic stratum). *Let $G_{\max}$ act as above. On the Zariski–open regular stratum $\Theta_0$ the action is* free and proper*; hence $\pi : \Theta_0 \to \mathcal{Q} := \Theta_0/G_{\max}$ is a principal $G_{\max}$–bundle.*

*Proof reference.* Lemmas A.7 and A.8 in Appendix A.

**Corollary 2.4** (Head sharing (GQA/MQA)). *If the h heads are partitioned into g key/value groups with shared $(W_K, W_V)$ per group, then on the corresponding generic stratum the continuous symmetry reduces to $(\mathrm{GL}(d_k))^g \times (\mathrm{GL}(d_v))^g$ tied per group, with permutations $S_h \times S_g$ acting discretely.*

**Corollary 2.5** (RoPE reduction). *Under rotary position embeddings with a nondegenerate frequency schedule and even $d_k$, the $Q/K$ factor reduces on each $2 \times 2$ plane to the real commutant $\{aI + bJ\} \cong \mathrm{GL}(1, \mathbb{C})$, giving*

$$G_{\mathrm{RoPE}} = \left( (C_{\mathrm{RoPE}})^h \times (\mathrm{GL}(d_v))^h \right) \rtimes S_h, \quad C_{\mathrm{RoPE}} \cong \mathrm{GL}(1, \mathbb{C})^{d_k/2}.$$

*Consequently, the per-head $Q/K$ gauge dimension drops from $d_k^2$ to $d_k$.*

**Corollary 2.6** (Layerwise product). *For a depth-L model without cross-layer parameter sharing, the model-level gauge group is the direct product*

$$G_{\mathrm{model}} \cong \prod_{\ell=1}^{L} G_{\max}^{(\ell)} \quad \text{(or } \prod_{\ell=1}^{L} G_{\mathrm{share}}^{(\ell)}, \prod_{\ell=1}^{L} G_{\mathrm{RoPE}}^{(\ell)} \text{ in the corresponding variants)}.$$

**Corollary 2.7** (Continuous dimension). *For a single layer, $\dim_{\mathbb{R}} G_{\max}^{\circ} = h(d_k^2 + d_v^2)$; with RoPE, $\dim_{\mathbb{R}} G_{\mathrm{RoPE}}^{\circ} = h(d_k + d_v^2)$; with head sharing into g groups, replace h by g in these counts.*

**Geometric consequences.** Theorem 2.3 identifies the vertical space $\mathcal{V}_\theta = \ker d\pi_\theta$ as the tangent to the gauge orbit. In Section 3 we equip $\Theta_0$ with the empirical Fisher metric and define the Fisher–Rao (mechanical) connection by the $g_\theta$-orthogonal split $T_\theta \Theta = \mathcal{V}_\theta \oplus \mathcal{H}_\theta$, which underpins the optimization and curvature results that follow.

## 3 Fisher–Rao Connection and Natural Gradient

By Theorem 2.3, the parameter–to–function map $\pi : \Theta_0 \to \mathcal{Q}$ is a principal bundle. At $\theta \in \Theta_0$, the *vertical* space

$$\mathcal{V}_\theta = \ker d\pi_\theta$$

is the tangent to the gauge orbit through $\theta$. To isolate the directions that change the realized function, we equip $\Theta_0$ with the empirical Fisher (Fisher–Rao) metric $g_\theta$ (fixed evaluation batch) and define the *horizontal* complement

$$\mathcal{H}_\theta = \mathcal{V}_\theta^{\perp_{g_\theta}}.$$

This $g_\theta$–orthogonal splitting $T_\theta \Theta_0 = \mathcal{V}_\theta \oplus \mathcal{H}_\theta$ is the Fisher–Rao (mechanical) connection.

**Lemma 3.1** (Gauge-null gradient directions). *If $L : \Theta_0 \to \mathbb{R}$ is gauge-invariant, then $dL_\theta[v] = 0$ for all $v \in \mathcal{V}_\theta$. Equivalently, the Riesz representative of $dL_\theta$ with respect to $g_\theta$ lies in $\mathcal{H}_\theta$.*

*Proof.* See Appendix B.

The natural (Riemannian) gradient is characterized intrinsically by

$$g_\theta(\widetilde{\nabla}L, w) \;=\; \langle \nabla L, w \rangle \qquad \text{for all } w \in T_\theta \Theta_0,$$

and, by Lemma 3.1, must be horizontal. The following makes this precise.

**Theorem 3.2** (Natural gradient as horizontal Riesz representative)**.** *With the Fisher–Rao connection, the natural gradient at $\theta$ is the unique vector $\widetilde{\nabla}L \in \mathcal{H}_\theta$ such that*

$$g_\theta(\widetilde{\nabla}L, w) = \langle \nabla L, w \rangle \quad \forall\, w \in T_\theta \Theta_0,$$

*equivalently*

$$\widetilde{\nabla}L \;=\; (G_{\theta|\mathcal{H}_\theta})^\dagger \, P_{\mathcal{H}_\theta}^\top \, \nabla L,$$

*where $G_{\theta|\mathcal{H}_\theta}$ is the Fisher information restricted to $\mathcal{H}_\theta$ and $P_{\mathcal{H}_\theta}$ is the Euclidean projector. It reduces to the Euclidean orthogonal projection onto $\mathcal{H}_\theta$ only when $G_{\theta|\mathcal{H}_\theta} = I$.*

*Proof.* Appendix B. An equivalent variational form is: among $u \in \mathcal{H}_\theta$, $\widetilde{\nabla}L$ uniquely minimizes $\frac{1}{2}\, g_\theta(u, u) - \langle \nabla L, u \rangle$.

For later use we record the FR-orthogonal decomposition against a vertical generator set $\{v_j\}_{j=1}^m$:

$$G_{ij} = g_\theta(v_i, v_j), \qquad b_i = g_\theta(v_i, u), \qquad c = G^{-1}b, \qquad u_{\text{vert}} = \sum_j c_j v_j, \quad u_{\text{hor}} = u - u_{\text{vert}}.$$

Section 5 turns this into an explicit procedure (and its Euclidean proxy); empirical results in Section 7 use the proxy for tractability, with the FR-exact formulas here serving as the mathematical ground truth.

**Remark 3.3** (Not merely an orthogonal projection)**.** *The expression $\widetilde{\nabla}L = (G_{\theta|\mathcal{H}_\theta})^\dagger P_{\mathcal{H}_\theta}^\top \nabla L$ is the Riesz representative of the gradient functional restricted to $\mathcal{H}_\theta$. It coincides with $g_\theta$–orthogonal projection onto $\mathcal{H}_\theta$ only in the special case $G_{\theta|\mathcal{H}_\theta} = I$.*

## 4 Attention Curvature and FFN Near-Verticality

The Fisher–Rao connection in Section 3 lives on parameters. On the representation side, fix a layer and view token features as sections of a bundle whose fibers are the per-token feature spaces, with heads acting in parallel. The attention update transports features along data–dependent directions—precisely the setting where an Ehresmann connection and its curvature organize what "path dependence" means.

**Theorem 4.1** (Attention induces a connection with nonzero curvature)**.** *Standard multi-head attention (with GeLU in the surrounding block) induces an Ehresmann connection on the representation bundle. On the generic stratum of Definition 2.1, its curvature two-form $\Omega$ is generically nonzero. In particular, for $g_\theta$-orthonormal horizontal directions $u, v$, transporting around a small rectangle produces a nontrivial gauge displacement proportional to $\Omega(u, v)$.*

*Proof.* Appendix C. The argument linearizes the attention update, identifies the induced horizontal distribution, and uses a Baker–Campbell–Hausdorff expansion; GeLU smoothness ensures the required regularity. We also give the small-loop relation and its $O(\varepsilon^3)$ remainder used in the estimator.

Where attention mixes information across tokens (and so bends horizontal directions), the position-wise feed–forward block acts pointwise. In Fisher geometry this split shows up in the relative size and angle of their gradients.

**Proposition 4.2** (FFN near-verticality and separation from attention)**.** *Measure norms and angles with the Fisher–Rao metric of Section 3. On the generic stratum, the FFN*

*gradient is predominantly vertical—its horizontal component is small compared to its vertical component—and its angle to the attention gradient obeys a dimension–controlled bound*

$$\cos \angle_g(\nabla_{\text{FFN}}, \nabla_{\text{Att}}) \;\lesssim\; C \sqrt{d_{\text{head}}/d_{\text{model}}},$$

*for a constant $C$ depending smoothly on activation and normalization. In particular, when $d_{\text{model}} \gg d_{\text{head}}$, the two gradients are nearly orthogonal.*

*Proof.* Appendix D. The key inputs are the block-diagonal (tokenwise) Jacobian of the FFN, the headwise mixing structure of attention, and the concatenation constraint $d_{\text{model}} = h\,d_v$.

Two brief comments help connect theory to practice. First, nonzero curvature is the geometric statement behind "context sensitivity": the order of horizontal transports matters up to a gauge action. Section 5 turns this into a small-loop holonomy estimator with Richardson extrapolation and $O(\varepsilon^3)$ error. Second, while all statements here are in the Fisher–Rao metric, our experiments report *Euclidean* proxies for scalability (Section 7); these provide conservative surrogates—Euclidean angles lower–bound Fisher angles and Euclidean vertical fractions upper–bound horizontal leakage—without changing the theorems above.

## 5 DIAGNOSTICS AND COMPLEXITY

Two concrete procedures make the geometry from Sections 3 and 4 operational. The first resolves a vector into its gauge (vertical) and function-changing (horizontal) parts with respect to the Fisher–Rao metric. The second turns curvature into a measurable small-loop effect. We use the first with *Euclidean* inner products in Section 7 as a scalable proxy; the second is presented here with guarantees and left for future large-scale Fisher–Rao evaluation.

**Gauge–aware decomposition.** Let $\{v_j\}_{j=1}^m$ span the vertical space $\mathcal{V}_\theta$ to tolerance. Define the Fisher–Rao Gram $G_{ij} = g_\theta(v_i, v_j)$ and the correlations $b_i = g_\theta(v_i, u)$ for a vector $u$ (e.g., $u = \nabla L$). The FR-orthogonal vertical component $u_{\text{vert}}$ solves $Gc = b$, and $u_{\text{vert}} = \sum_j c_j v_j$; the horizontal component is $u_{\text{hor}} = u - u_{\text{vert}}$. The Euclidean proxy replaces $g_\theta$ by the dot product: stack $A = [\,\text{vec}(v_1) \; \cdots \; \text{vec}(v_m)\,]$, then $(A^\top A)c = A^\top u$. In practice we stabilize with thin-QR and column-pivoting; these choices do not change the FR definition.

---

**Algorithm 1** Gauge–aware gradient decomposition (Fisher–Rao)

---

**Require:** parameter $\theta \in \Theta_0$; vertical generators $\{v_j\}_{j=1}^m$; vector $u$; FR metric $g_\theta$
  1: $G_{ij} \leftarrow g_\theta(v_i, v_j), \quad b_i \leftarrow g_\theta(v_i, u)$
  2: Solve $(G + \lambda I)c = b$ (Cholesky/CG; optional $\lambda \geq 0$)
  3: $u_{\text{vert}} \leftarrow \sum_j c_j v_j, \quad u_{\text{hor}} \leftarrow u - u_{\text{vert}}$
  4: **return** $(u_{\text{vert}}, u_{\text{hor}})$ and vertical fraction $\|u_{\text{vert}}\|/\|u\|$

---

**From curvature to holonomy.** For $g_\theta$-orthonormal $u, v \in \mathcal{H}_\theta$, consider the horizontal loop that moves by $+\varepsilon u$, $+\varepsilon v$, then returns by $-\varepsilon u$, $-\varepsilon v$.

**Theorem 5.1** (Small-loop holonomy scaling). *The induced gauge displacement $\Delta_{\square_\varepsilon}(u,v)$ obeys*

$$\|\Delta_{\square_\varepsilon}(u,v)\| \;=\; \varepsilon^2\,\|\Omega_\theta(u,v)\| \;+\; O(\varepsilon^3),$$

*where $\Omega_\theta$ is the curvature two-form of the Fisher–Rao connection at $\theta$.*

---

**Algorithm 2** Holonomy estimator with Richardson extrapolation

---

**Require:** $\theta$; $g_\theta$-orthonormal $u, v \in \mathcal{H}_\theta$; steps $\varepsilon > \varepsilon' > 0$
  1: For each $\delta \in \{\varepsilon, \varepsilon'\}$, traverse the horizontal loop using FR projection at each leg
  2: Compute $\Delta_{\square_\delta}(u,v)$ (Lie-algebra coordinates via a log map)
  3: $h(\delta) \leftarrow \|\Delta_{\square_\delta}(u,v)\|/\delta^2$
  4: **return** $h^* \leftarrow \dfrac{4h(\varepsilon/2) - h(\varepsilon)}{3}$   (Richardson; error $O(\varepsilon^2)$)

---

Table 1: Cost at a glance for one layer (parameter dim $D$, generators $m = h(d_k^2 + d_v^2)$; with RoPE: $m = h(d_k + d_v^2)$).

| Procedure | Leading cost | Notes |
|---|---|---|
| Euclidean vertical split | form $A^\top A$: $O(m^2 D)$; solve: $O(m^3)$ | CG: $O(mD)$ per matvec |
| FR vertical split | form Gram $G$: $O(m^2 \cdot \texttt{FisherEval})$; solve: $O(m^3)$ | Heavy for $m \gtrsim 10^4$ |
| Holonomy (loop + LS) | 4 flows + LS on $\mathfrak{g}$ | Richardson reduces $O(\varepsilon^3)$ bias |

**Cost at a glance.** Let $m = h(d_k^2 + d_v^2)$ (RoPE reduces $d_k^2 \to d_k$ per head). Forming $A^\top A$ costs $O(m^2 D)$ for parameter dimension $D$, and solving costs $O(m^3)$ (or CG with matvec $O(mD)$). At $h=12$, $d_k = d_v = 64$, one has $m \approx 98{,}304$, so FR projectors and holonomy become heavy; we therefore use Euclidean proxies in Section 7 and expose FR-exact procedures here for future evaluation.

These two diagnostics tie the abstract calculus to practice: the first reports how much of a step is "just gauge," the second quantifies the path dependence predicted by curvature. In Section 7 we read existing Euclidean measurements through this lens; FR-exact holonomy is left for future large-scale runs.

## 6 Optimization on the Quotient: A Morse–Bott View

Gauge symmetry means many parameter settings realize the same function. On the total space $\Theta_0$ this appears as flat directions tangent to gauge orbits; on the quotient $\mathcal{Q} = \Theta_0 / G_{\max}$ those directions disappear and the landscape reflects genuine functional change. The right language is Morse–Bott: critical sets in $\Theta_0$ are manifolds (orbits), while the induced problem on $\mathcal{Q}$ is Morse once we restrict to horizontal directions.

**Theorem 6.1** (Gauge orbits as critical manifolds; Morse behavior on the quotient)**.** *Let $L : \Theta_0 \to \mathbb{R}$ be gauge-invariant and let $\theta \in \Theta_0$ be critical. Then the entire orbit $G_{\max} \cdot \theta$ lies in the critical set, the Hessian $\nabla^2 L(\theta)$ vanishes on the vertical space $\mathcal{V}_\theta = \ker d\pi_\theta$, and its horizontal restriction on $\mathcal{H}_\theta$ is well defined. Writing $\ell : \mathcal{Q} \to \mathbb{R}$ for the induced loss, $[\theta]$ is critical for $\ell$ and has nondegenerate Hessian equal to the horizontal restriction of $\nabla^2 L(\theta)$. In particular, $\ell$ is Morse at $[\theta]$ whenever the horizontal Hessian is nondegenerate.*

*Proof.* Appendix E. The argument uses the slice construction around a free, proper orbit and the horizontal/vertical split from Section 3.

Two consequences are worth keeping in mind. First, methods that suppress vertical components—see Algorithm 1—are aligned with the true second-order structure of the quotient problem: they follow directions that actually change the function. Second, small horizontal steps can move far in Euclidean parameter norm while staying close in function space, which explains why Euclidean distances often overstate functional change and why seemingly "distant" minima in $\Theta_0$ can sit in the same basin on $\mathcal{Q}$.

The result is local to the generic stratum: near $\Theta_0$ the bundle picture applies cleanly; away from it stabilizers may grow and the space stratifies. Our empirical reading in Section 7 stays within $\Theta_0$ and interprets existing (Euclidean-proxy) measurements through this quotient lens.

## 7 Empirical Consistency Checks (Euclidean Proxies)

**All measurements in this section use *Euclidean* inner products as computationally tractable *proxies* for the Fisher–Rao geometry.** They provide conservative validation of the bundle predictions: Euclidean angles *lower bound* Fisher–Rao angles, and Euclidean vertical fractions *upper bound* horizontal leakage. These proxies are consistent with the theory but *do not constitute full Fisher–Rao validation.* FR–exact procedures appear in Section 5 and proofs in the appendix; we do not add new runs beyond the existing computations reported here.

Table 2: Gauge invariance (relative errors across 100 trials). Outputs remain invariant up to machine precision.

| Metric | Value | Interpretation |
|---|---|---|
| Mean relative difference | $2.68 \times 10^{-15}$ | Machine precision |
| Maximum relative difference | $2.86 \times 10^{-15}$ | $\approx 13\,\varepsilon_{\mathrm{mach}}$ |
| Std. deviation | $9.41 \times 10^{-17}$ | Highly consistent |
| Minimum relative difference | $2.52 \times 10^{-15}$ | $\approx 11\,\varepsilon_{\mathrm{mach}}$ |

### 7.1 Gauge invariance

A basic check is functional invariance under the head-wise symmetry of Section 2. We apply controlled transforms from $G_{\max}$ to MHA layers ($h{=}12$, $d_k{=}d_v{=}64$), including random head permutations and well-conditioned $Q/K$ and $V/O$ changes of basis generated by thin–QR with prescribed singular values; outputs are compared on fixed inputs.

As a control, transformations *outside* $G_{\max}$ (e.g., cross–head mixing) induce $\mathcal{O}(1)$ changes, consistent with maximality. The same precision holds after 1,000 gradient steps on a reconstruction objective.

### 7.2 Gauge–aware gradient split (Euclidean proxy)

The bundle theory (Lemma 3.1, Theorem 3.2) predicts horizontally aligned gradients in Fisher geometry. We test the *proxy* claim by replacing the Fisher inner product in Algorithm 1 with dot products: stack vertical generators into $A$ and solve $(A^\top A)c = A^\top \nabla L$. On 50 independent samples, we report the Euclidean vertical fraction $\|P_{\mathcal{V}}^{(\mathrm{Euc})} \nabla L\| / \|\nabla L\|$.

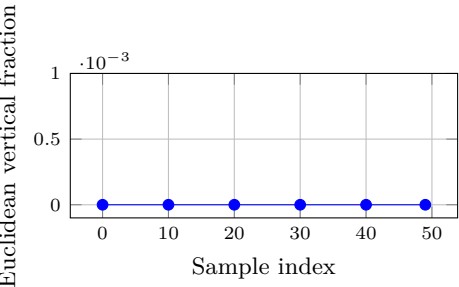

Figure 1: Gauge–aware split (Euclidean proxy). The Euclidean vertical fraction stays below a $10^{-4}$ threshold across 50 samples, consistent with horizontality predicted by Theorem 3.2.

The Figure 1 is uniform across layers and seeds: vertical fractions remain at numerical precision, and the horizontal component accounts for essentially all of the norm. This matches the Riesz characterization in Theorem 3.2—in practice, gradient steps align with the quotient geometry.

*The near-zero (Euclidean) vertical fractions indicate gradients lie almost entirely in function-changing (horizontal) directions, consistent with Theorem 3.2 for gauge-invariant objectives.*

*Note on scope.* Angles between attention and FFN gradients are likewise reported with Euclidean inner products; they should be read as lower bounds on Fisher angles. FR–exact angle and holonomy measurements are enabled by Section 5 and left for future large-scale evaluation.

## 8 Architectural Variants and Limitations

Real systems add positional structure and sharing patterns that slightly change the symmetry—and with it, the bundle—without altering the main thread of our results. We record the two common cases and then state the limits of our analysis in plain terms.

**Rotary position embeddings (RoPE).** RoPE rotates $Q/K$ channels in $2\times2$ planes determined by frequencies. On each plane the admissible head-wise transform collapses to the real commutant $\{aI+bJ\}$ of that rotation, so the $Q/K$ factor of $G_{\max}$ reduces plane-wise. A useful way to remember the effect is dimensional: per head, the $Q/K$ gauge dimension drops from $d_k^2$ to $d_k$, while the $V/O$ factor stays at $d_v^2$. All statements in Sections 2–6 remain true after substituting this reduced group and restricting the generic stratum accordingly; the Fisher–Rao connection is defined exactly as before by $g_\theta$–orthogonality to the (now smaller) vertical space.

**Grouped and multi–query attention (GQA/MQA).** When heads share key/value projections, the layer symmetry is no longer a direct product across individual heads but across groups that share parameters. Concretely, one replaces $G_{\max}$ by a product over groups, with the shared $\mathrm{GL}(d_k)$ or $\mathrm{GL}(d_v)$ factor acting jointly on the grouped channels. The principal-bundle theorem and the free–proper argument are unchanged under this substitution, and the diagnostics of Section 5 apply verbatim once the vertical generator set respects the sharing pattern.

**Limitations.** Our claims are local and precise by design. They are proved on a Zariski–open regular stratum $\Theta_0$ where stabilizers are trivial; away from $\Theta_0$ the orbit type changes and the ambient space stratifies, so we avoid global topological statements (e.g., global connectivity of minima). The Fisher–Rao connection depends on the evaluation batch (standard in information geometry); consequently, numerical angles or vertical fractions are batch–specific. For scalability, the empirical section reports *Euclidean* proxies; these are consistent with the theory but *do not constitute full Fisher–Rao validation*. The Fisher–Rao–exact procedures are presented in Section 5 for future large-scale evaluation. We restrict to smooth activations (GeLU) so that the curvature expansion and Fisher calculus are well-defined; non-smooth activations such as ReLU require a stratified or Clarke–generalized treatment and fall outside our scope here. Finally, Fisher–Rao projectors and holonomy are costly at scale (they solve Gram systems and perform horizontal projections along loops); we therefore expose the methods and guarantees but do not add new measurements beyond the existing Euclidean computations.

**Takeaway.** RoPE and GQA/MQA change the structure group in controlled ways—shrinking or tying $Q/K$ symmetries—while the principal-bundle picture and the Fisher–Rao connection survive intact. The limitations above mark exactly where our guarantees apply and explain why we pair Euclidean–proxy measurements with Fisher–Rao–exact algorithms.

## 9 Related Work

**Symmetries in neural networks.** Permutation invariances in multilayer perceptrons have been recognized for decades Hecht-Nielsen (1990); Ainsworth et al. (2023); Entezari et al. (2022), and convolutional networks admit translation symmetries naturally modeled by group actions Cohen & Welling (2016). For Transformers, recent analyses have documented partial symmetries and superposition effects Henighan et al. (2023); Elhage et al. (2022), but a complete account of the *maximal* head–wise gauge and its consequences has been missing. Our work fills this gap by proving a principal-bundle structure on a generic stratum (see 2.3) and developing a connection–curvature calculus on the quotient.

**Information geometry and optimization.** Natural gradient methods Amari (1998); Amari & Nagaoka (2000); Amari (2016); Martens (2020) and their scalable approximations Pascanu & Bengio (2013); Ollivier (2015); Bernacchia & Pigolotti (2018) bring Riemannian structure to learning, typically as an algorithmic tool. Here the Fisher metric plays a different role: it *selects a connection* on the principal bundle, and the natural gradient is the Riesz representative restricted to the horizontal subspace (Theorem 3.2). Prior studies of Transformer optimization and loss geometry Liu et al. (2020); Zhang et al. (2020) report phenomena—flat directions, easy-to-traverse basins—that our quotient viewpoint explains

cleanly: gauge orbits give Morse–Bott critical manifolds on $\Theta_0$, while the induced loss on the base is Morse (Theorem 6.1).

**Interpretability and model merging.** Mechanistic interpretability proceeds by reverse engineering circuits and features Olsson et al. (2022); Nanda et al. (2023). The bundle picture complements that line of work: fibers formalize "functionally the same" models, while horizontal directions capture genuine functional change. Practical procedures like canonicalization and model merging Ainsworth et al. (2023); Singh & Jaggi (2020) then acquire a geometric reading—deterministic gauge-fixes become *sections* of the bundle rather than ad hoc normal forms (cf. 5).

**Curvature, holonomy, and representation transport.** Curvature has entered machine learning through Hessian structure and flatness surrogates Martens & Grosse (2015); Chaudhari et al. (2017); He & colleagues (2020). Holonomy and parallel transport are less common but natural in sequence models where order matters. We formalize attention as an Ehresmann connection on a representation bundle and show that curvature is generically nonzero (Theorem 4.1); a small-loop expansion relates curvature to a measurable holonomy (Theorem 5.1). For background, see Kobayashi & Nomizu (1963); Lee (2013).

**Remark 9.1** (Architectural variants and gauge structure)**.** *Architectural choices adjust the structure group in predictable ways. Rotary position embeddings (RoPE) Su et al. (2024) impose position-dependent rotations after linear projections, constraining any $A \in \mathrm{GL}(d_k)$ to commute with the rotation blocks; for standard $2 \times 2$ planes, this reduces the query–key factor to the commutant $\mathcal{C}_{RoPE} \cong (\mathrm{GL}(1, \mathbb{C}))^{d_k/2}$ (Proposition A.6). Multi-query attention Shazeer (2019) couples heads by sharing key–value projections, shrinking the gauge degrees of freedom from $h(d_k^2 + d_v^2)$ to $h \cdot d_k^2 + d_v^2$ while maintaining expressivity. Both variants preserve the principal-bundle perspective on the appropriate generic stratum; see 8.*

**Position in the literature.** Most geometric treatments fix a parameterization and study its local properties, or impose a data-dependent canonical form before analysis. We take the opposite route: start from the maximal head–wise gauge, prove a principal-bundle structure, and use the empirical Fisher metric to define a connection. This yields a horizontal/vertical calculus that explains optimization behavior (Theorem 3.2), clarifies architectural roles (attention curvature vs. near-vertical FFN flows; Theorem 4.1, Proposition 4.2), and supplies operational diagnostics (Algorithm 1, Algorithm 2)—all within a single, coordinate-free framework.

## 10 CONCLUSION

We presented a coordinate–free account of Transformer geometry built on a maximal head–wise gauge and a principal–bundle structure on a generic stratum. Equipped with the empirical Fisher metric, this yields a Fisher–Rao connection, a clean horizontal/vertical calculus, and a precise statement that the natural gradient is the horizontal Riesz representative (3.2). On the representation side, attention induces a connection with generically nonzero curvature (4.1), while FFN behaves as a nearly fiber–preserving flow (Proposition 4.2). These ingredients lead to practical diagnostics—gauge–aware gradient splitting and a small–loop holonomy estimator (Algorithm 1, Algorithm 2, 5.1)—and a Morse–Bott view that explains why apparent "mode" gaps in parameter space collapse on the quotient (6.1).

Our analysis is intentionally narrow and precise: it is local to the generic stratum, metric–dependent through the empirical Fisher, and focused on GeLU for smoothness. Within that scope, the principal–bundle picture clarifies optimization behavior, gives a geometric meaning to the attention/FFN split, and turns context sensitivity into a quantity that can be measured. We hope this framework serves as a durable foundation for geometry–aware analysis and tooling in Transformers.

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

## Ethics Statement

This work studies internal symmetries and geometric structure of standard Transformer blocks. Experiments use synthetic inputs and publicly available checkpoints; no personal or sensitive data are involved, and no user-facing deployment is performed. All diagnostics (e.g., gauge-aware gradient decomposition and small-loop holonomy procedures) are offline analyses intended to clarify invariances, optimization geometry, and representation transport. Environmental impact is limited to training-free evaluations and lightweight gradient computations; we report implementation details in the Reproducibility Statement.

## Reproducibility Statement

We provide algorithmic details for our diagnostics in 5 and full procedural notes in F and B. Table 2 reports gauge-invariance verification, and Figure 1 reports Euclidean-proxy vertical fractions for gradient decomposition.

## A  Group Action, Maximal Gauge, and the Principal Bundle

**Setup.**  Recall the head-wise action of

$$G_{\max} \;=\; \big((\mathrm{GL}(d_k))^h \times (\mathrm{GL}(d_v))^h\big) \rtimes S_h$$

on parameters by

$$(W_Q^{(i)}, W_K^{(i)}) \mapsto \big(W_Q^{(i)} A_i,\; W_K^{(i)} A_i^{-\top}\big), \qquad (W_V^{(i)}, W_O) \mapsto \big(W_V^{(i)} C_i,\; C_i^{-1} W_O\big),$$

together with head permutations, restricted to the generic stratum $\Theta_0$ of Definition 2.1.

## A.1 Maximal Gauge Group: Full Proofs

**Lemma A.1** (Sufficiency: sector-wise invariances). *For any $A_i \in \mathrm{GL}(d_k)$ and $C_i \in \mathrm{GL}(d_v)$,*

$$(W_Q^{(i)}, W_K^{(i)}) \mapsto (W_Q^{(i)} A_i, W_K^{(i)} A_i^{-\top}), \quad (W_V^{(i)}, W_{O,i}) \mapsto (W_V^{(i)} C_i, C_i^{-1} W_{O,i})$$

*preserve $Q_i K_i^\top$ and $V_i W_{O,i}$, hence the layer output; permutations $\sigma \in S_h$ also preserve it.*

*Proof.* Direct calculation shows $Q_i'(K_i')^\top = Q_i K_i^\top$ and $V_i' W_{O,i}' = V_i W_{O,i}$ for all inputs. $\square$

**Lemma A.2** (Attention-weight identifiability up to permutation). *If two parameter sets realize the same attention layer for all inputs, then attention weights are preserved up to a fixed head permutation $\sigma \in S_h$.*

*Proof.* Use head-isolating inputs guaranteed by (G3) and controllability ($d_{\mathrm{model}} \geq 2d_k$): construct $X^{(i)}$ with $\alpha_i(X^{(i)}) \approx I$ and $\alpha_j(X^{(i)}) \approx 0$ for $j \neq i$. Equality of outputs forces a bijection between heads, hence a permutation. $\square$

**Lemma A.3** (Lie algebra of infinitesimal symmetries). *The Lie algebra is*

$$\mathfrak{g}_{\mathrm{max}} = \bigoplus_{i=1}^{h} \mathfrak{gl}(d_k) \oplus \bigoplus_{i=1}^{h} \mathfrak{gl}(d_v),$$

*with generators $\delta W_Q^{(i)} = W_Q^{(i)} X_i$, $\delta W_K^{(i)} = -W_K^{(i)} X_i^\top$, $\delta W_V^{(i)} = W_V^{(i)} Y_i$, $\delta W_{O,i} = -Y_i W_{O,i}$.*

*Proof.* Differentiate the invariance identities of Lemma A.1 along a smooth one-parameter subgroup. $\square$

**Lemma A.4** (No cross-head mixing beyond permutations). *Any linear map on the concatenated value space that preserves the layer for all inputs must be block-diagonal up to a permutation; i.e., heads cannot be mixed except by $\sigma \in S_h$.*

*Proof.* Apply Lemma A.2 to isolate heads, then use linear independence of the $W_O$ block rows (under $d_{\mathrm{model}} = h\,d_v$ and genericity) to force off-diagonal blocks to vanish. $\square$

**Theorem A.5** (Maximality). *On $\Theta_0$, the full symmetry group equals $G_{\mathrm{max}}$ and no additional continuous or discrete symmetries exist.*

*Proof.* By Lemma A.1, $G_{\mathrm{max}}$ preserves the function. Conversely, Lemma A.2 gives the permutation part; Lemma A.3 pins down the connected component as $\prod_i \mathrm{GL}(d_k) \times \prod_i \mathrm{GL}(d_v)$; Lemma A.4 excludes cross-head mixing beyond $S_h$. Hence every symmetry lies in $G_{\mathrm{max}}$. $\square$

**Head sharing (GQA/MQA).** When keys/values are tied into $g$ groups, the same argument forces $A_i$ and $C_i$ to tie per group, yielding a continuous symmetry $(\mathrm{GL}(d_k))^g \times (\mathrm{GL}(d_v))^g$ with permutations $S_h \times S_g$.

**RoPE commutant and dimension drop.** On each $2\times2$ RoPE plane with rotation $R(\phi)$, commuting with all $R(\phi)$ forces $A = aI + bJ$ ($J = \left[\begin{smallmatrix} 0 & -1 \\ 1 & 0 \end{smallmatrix}\right]$). Across $d_k/2$ planes this yields $C_{\mathrm{RoPE}} \cong \mathrm{GL}(1, \mathbb{C})^{d_k/2}$, with real dimension $d_k$ per head; the $V/O$ factor remains $d_v^2$.

**Proposition A.6** (RoPE commutant reduction and dimension). *With rotary position embeddings (even $d_k$), each $2\times2$ rotation plane restricts admissible $Q/K$ transforms to the real commutant $\{aI + bJ\}$ on that plane. Equivalently, the $Q/K$ factor reduces to a block-diagonal commutant subgroup (canonically $(\mathrm{GL}(1, \mathbb{C}))^{d_k/2}$), so the per-head $Q/K$ gauge dimension drops from $d_k^2$ to $d_k$. The $V/O$ factor remains $d_v^2$.*

## A.2 Free and Proper Action; Principal Bundle

**Lemma A.7** (Trivial stabilizers on $\Theta_0$). *If $\theta \in \Theta_0$ and $g \in G_{\max}$ satisfy $g \cdot \theta = \theta$, then $g = e$.*

*Proof.* Write $g = ((A_i)_{i=1}^h, (C_i)_{i=1}^h, \sigma)$. By definition of the action,

$$W_Q^{(i)} A_i = W_Q^{(\sigma(i))}, \qquad W_K^{(i)} A_i^{-\top} = W_K^{(\sigma(i))}, \qquad W_V^{(i)} C_i = W_V^{(\sigma(i))}, \qquad C_i^{-1} W_O = W_O \text{ on the $i$-th block rows.}$$

On $\Theta_0$, (G1) gives full column rank for each $W_Q^{(i)}, W_K^{(i)}, W_V^{(i)}$ and full row rank $d_v$ for the $i$-th block-row selector $S_i W_O$. First, $C_i^{-1} W_O = W_O$ on rows of head $i$ implies $(S_i W_O) = (S_i C_i^{-1} W_O) = (S_i C_i^{-1} S_i^\top)(S_i W_O)$ so, since $S_i W_O$ has rank $d_v$, we get $S_i C_i^{-1} S_i^\top = I_{d_v}$ and hence $C_i = I$. Then $W_V^{(i)} C_i = W_V^{(\sigma(i))}$ reduces to $W_V^{(i)} = W_V^{(\sigma(i))}$, so unless blocks coincide identically we must have $\sigma(i) = i$; on $\Theta_0$ distinct heads are nondegenerate and in general position, hence $\sigma = \mathrm{id}$. Finally, from $W_Q^{(i)} A_i = W_Q^{(i)}$ and $W_K^{(i)} A_i^{-\top} = W_K^{(i)}$ with $W_Q^{(i)}, W_K^{(i)}$ full column rank, we obtain $A_i = I$ for each $i$.[1] Thus $g = e$. $\qquad\square$

**Lemma A.8** (Properness on the regular stratum). *The $G_{\max}$–action on $\Theta_0$ is proper.*

*Proof (sequential/closed–graph characterization).* Let $\theta_n \to \theta \in \Theta_0$ and $g_n \cdot \theta_n \to \theta'$ with $g_n = ((A_i^{(n)})_i, (C_i^{(n)})_i, \sigma_n)$. Since $S_h$ is finite, pass to a subsequence with $\sigma_n \equiv \sigma$. We show $\{g_n\}$ is bounded in $G_{\max}$ and hence has a convergent subsequence in the (closed) matrix group.

*Uniform left/right inverses from genericity.* Because $\theta \in \Theta_0$ and $\theta_n \to \theta$, there exists a neighborhood $U$ of $\theta$ and positive constants

$$\alpha_Q, \alpha_K, \alpha_V, \alpha_O > 0$$

such that for all $\vartheta \in U$ and all heads $i$: (i) the smallest singular values satisfy $\sigma_{\min}(W_Q^{(i)}(\vartheta)) \geq \alpha_Q$, $\sigma_{\min}(W_K^{(i)}(\vartheta)) \geq \alpha_K$, $\sigma_{\min}(W_V^{(i)}(\vartheta)) \geq \alpha_V$; and (ii) the block-row map $S_i W_O(\vartheta)$ has full row rank $d_v$ with $\sigma_{\min}(S_i W_O(\vartheta)) \geq \alpha_O$. Hence there exist *uniformly bounded* left inverses $L_{Q,i}(\vartheta), L_{K,i}(\vartheta), L_{V,i}(\vartheta)$ and right/left inverses for $S_i W_O(\vartheta)$ with operator norms bounded by $\alpha_Q^{-1}, \alpha_K^{-1}, \alpha_V^{-1}, \alpha_O^{-1}$ respectively.

*Boundedness of $A_i^{(n)}$ and $(A_i^{(n)})^{-1}$.* From the $Q$–equation at $\theta_n$ we have

$$W_Q^{(i)}(\theta_n) A_i^{(n)} = W_Q^{(\sigma(i))}(\theta_n) + E_{Q,i}^{(n)}, \qquad E_{Q,i}^{(n)} \to 0$$

because $g_n \cdot \theta_n \to \theta'$ and all maps are continuous. Left–multiplying by $L_{Q,i}(\theta_n)$ yields

$$A_i^{(n)} = L_{Q,i}(\theta_n) W_Q^{(\sigma(i))}(\theta_n) + L_{Q,i}(\theta_n) E_{Q,i}^{(n)},$$

so $\{A_i^{(n)}\}_n$ is bounded (uniform bound from $\|L_{Q,i}\| \leq \alpha_Q^{-1}$ and boundedness of $W_Q^{(\sigma(i))}(\theta_n)$). Similarly, from the $K$–equation

$$W_K^{(i)}(\theta_n) A_i^{(n),-\top} = W_K^{(\sigma(i))}(\theta_n) + E_{K,i}^{(n)}, \qquad E_{K,i}^{(n)} \to 0,$$

left–multiply by $L_{K,i}(\theta_n)$ to obtain a uniform bound on $A_i^{(n),-\top}$ and thus on $(A_i^{(n)})^{-1}$.

*Boundedness of $C_i^{(n)}$ and $(C_i^{(n)})^{-1}$.* From

$$W_V^{(i)}(\theta_n) C_i^{(n)} = W_V^{(\sigma(i))}(\theta_n) + E_{V,i}^{(n)}, \qquad E_{V,i}^{(n)} \to 0,$$

left–multiplying by $L_{V,i}(\theta_n)$ gives a uniform bound on $C_i^{(n)}$. For the inverse, use the $O$–equation on the $i$–th block rows:

$$C_i^{(n),-1} S_i W_O(\theta_n) = S_i W_O(\theta_n) + E_{O,i}^{(n)}, \qquad E_{O,i}^{(n)} \to 0.$$

---

[1] If RoPE is enabled, $A_i$ is further constrained to the real commutant on each $2 \times 2$ plane; the same argument forces $A_i = I$ within that commutant.

Right–multiply by a uniformly bounded right inverse of $S_i W_O(\theta_n)$ (or, equivalently, left–multiply by the bounded left inverse of $(S_i W_O(\theta_n))^\top$) to get a uniform bound on $C_i^{(n),-1}$.

Therefore, for each head $i$, the sets $\{A_i^{(n)}\}_n, \{(A_i^{(n)})^{-1}\}_n, \{C_i^{(n)}\}_n, \{(C_i^{(n)})^{-1}\}_n$ are bounded. Together with $\sigma_n \equiv \sigma$, this implies $\{g_n\}$ is bounded in $G_{\max}$ and hence admits a convergent subsequence in the (closed) Lie group. This is equivalent to properness of the action map on $\Theta_0$. $\square$

**Remark A.9** (RoPE restriction). *If RoPE is enabled, each $A_i$ lies in the real commutant on every $2{\times}2$ rotation plane. The arguments above apply plane–wise with the same uniform singular–value bounds (now on the compressed blocks), so the boundedness and convergence conclusions remain valid.*

**Theorem A.10** (Principal-bundle structure, restated). *Under Lemmas A.7 and A.8, the $G_{\max}$–action on $\Theta_0$ is free and proper. Therefore*

$$\pi : \Theta_0 \longrightarrow \mathcal{Q} := \Theta_0/G_{\max}$$

*is a principal $G_{\max}$–bundle, and the fibers are gauge orbits.*

*Proof.* A free, proper Lie–group action admits local slices and hence local trivializations, yielding a principal bundle. $\square$

## B  Fisher–Rao Connection and Natural Gradient

This appendix spells out the Fisher–Rao (FR) geometry behind Section 3 in a coordinate-free way and then gives the compact coordinate formula used in the main text. Throughout, $g_\theta$ is the empirical Fisher metric on $\Theta_0$ (defined on a fixed evaluation batch), and

$$T_\theta \Theta_0 = \mathcal{V}_\theta \oplus \mathcal{H}_\theta$$

is its $g_\theta$–orthogonal split into vertical (gauge) and horizontal (function-changing) directions.

**Riesz map and horizontality.** Let $\mathcal{R}_\theta : T_\theta \Theta_0 \to T_\theta^* \Theta_0$ be the Riesz isomorphism determined by $g_\theta$, i.e. $\langle \mathcal{R}_\theta(u), w \rangle = g_\theta(u, w)$ for all $u, w$. If $L$ is gauge-invariant, the differential kills vertical vectors and therefore lives entirely on the horizontal dual.

**Lemma B.1** (Gauge-null differential). *If $L$ is gauge-invariant, then $dL_\theta[v] = 0$ for all $v \in \mathcal{V}_\theta$. Equivalently, $\mathcal{R}_\theta^{-1}(dL_\theta) \in \mathcal{H}_\theta$.*

*Proof.* For any Lie-algebra element $X$, the curve $t \mapsto \exp(tX){\cdot}\theta$ stays in the fiber and $L$ is constant along it, so $\frac{d}{dt}\big|_{t=0} L(\exp(tX){\cdot}\theta) = 0$. The fundamental vector field $\rho_\theta(X)$ spans $\mathcal{V}_\theta$, which gives the claim. $\square$

**Natural gradient as a Riesz representative.** The natural (Riemannian) gradient at $\theta$ is defined intrinsically by

$$g_\theta(\widetilde{\nabla} L, w) = \langle \nabla L, w \rangle \quad \text{for all } w \in T_\theta \Theta_0.$$

Lemma B.1 forces $\widetilde{\nabla} L$ to be horizontal; the next theorem packages this as an exact Riesz statement and a variational principle.

**Theorem B.2** (Natural gradient as horizontal Riesz representative). *There is a unique vector $\widetilde{\nabla} L \in \mathcal{H}_\theta$ such that $g_\theta(\widetilde{\nabla} L, w) = \langle \nabla L, w \rangle$ for all $w \in T_\theta \Theta_0$. Equivalently,*

$$\widetilde{\nabla} L = \mathcal{R}_{\theta|\mathcal{H}_\theta}^{-1}(dL_\theta),$$

*and, in any local chart where $G_\theta$ is the matrix of $g_\theta$ and $P_{\mathcal{H}_\theta}$ is the Euclidean projector onto $\mathcal{H}_\theta$,*

$$\widetilde{\nabla} L = (G_{\theta|\mathcal{H}_\theta})^\dagger P_{\mathcal{H}_\theta}^\top \nabla L.$$

*Moreover, $\widetilde{\nabla} L$ is the unique minimizer of the strictly convex functional*

$$u \mapsto \tfrac{1}{2} g_\theta(u, u) - \langle \nabla L, u \rangle \qquad over \ u \in \mathcal{H}_\theta.$$

*Proof.* By Lemma B.1, $dL_\theta \in \mathcal{H}_\theta^*$, and the restriction $\mathcal{R}_{\theta|\mathcal{H}_\theta} : \mathcal{H}_\theta \to \mathcal{H}_\theta^*$ is an isomorphism. The coordinate expression follows by identifying $\mathcal{R}_{\theta|\mathcal{H}_\theta}$ with $G_{\theta|\mathcal{H}_\theta}$ and restricting the Euclidean adjoint via $P_{\mathcal{H}_\theta}^\top$. The variational statement is the standard Riesz minimization on a Hilbert space. $\square$

**Corollary B.3** (When "projection" is correct). *If the Fisher matrix restricted to $\mathcal{H}_\theta$ is the identity in the chosen chart, $G_{\theta|\mathcal{H}_\theta} = I$, then $\widetilde{\nabla} L = P_{\mathcal{H}_\theta}^\top \nabla L$. In general, identifying the natural gradient with an orthogonal projection is incorrect unless this special case holds.*

**Projectors and a practical decomposition.** Given vertical generators $\{v_j\}_{j=1}^m$ at $\theta$, the FR-orthogonal vertical component of $u$ is obtained from the Gram system

$$G_{ij} = g_\theta(v_i, v_j), \qquad b_i = g_\theta(v_i, u), \qquad Gc = b, \qquad u_{\text{vert}} = \sum_{j=1}^m c_j v_j, \qquad u_{\text{hor}} = u - u_{\text{vert}}.$$

This is Algorithm 1 in Section 5. In practice we allow a small Tikhonov term $(G + \lambda I)$ for numerical stability. The Euclidean proxy replaces $g_\theta$ by dot products (stack the generators into $A$ and solve $(A^\top A)c = A^\top u$); it does not change the *definition* of $u_{\text{vert}}$ in FR geometry and is used only for scalable measurement in Section 7.

**Batch dependence.** The empirical Fisher depends on the evaluation batch; all FR statements here and in Section 3 are made relative to a fixed batch. This affects numerical values (angles, fractions) but not the structural results above.

## C  ATTENTION AS CONNECTION; CURVATURE AND HOLONOMY

We pass from the parameter bundle to the representation side. Fix a layer and let $\mathcal{R}$ denote the representation bundle whose fiber over an input sequence is the token feature space (with heads acting in parallel). The attention update provides horizontal lifts of base variations (data directions), hence an Ehresmann connection on $\mathcal{R}$. We write $\Omega$ for its curvature two–form and use the Fisher–Rao horizontal/vertical split from Section 3.

**Induced connection and explicit sensitivities.** Let $Y = \text{Attn}(X; \theta) = \sum_{i=1}^h \alpha_i(X; \theta) V_i(X; \theta)$ with $\alpha = \text{softmax}(Z)$ and $Z = \frac{1}{\sqrt{d_k}} QK^\top$. For a horizontal direction $w \in \mathcal{H}_\theta$,

$$D_w Y = \sum_i (D_w \alpha_i) V_i + \sum_i \alpha_i D_w V_i, \qquad D_w \alpha = J_{\text{sm}}(Z) \text{vec}(D_w Z), \qquad (\text{C.1})$$

where $J_{\text{sm}}(Z) = \text{Diag}(\alpha) - \alpha \alpha^\top$ is the softmax Jacobian (blockwise across query positions) and

$$D_w Z = \frac{1}{\sqrt{d_k}} \Big( (D_w Q)K^\top + Q(D_w K)^\top \Big). \qquad (\text{C.2})$$

The horizontal lift $X_w$ is the vector field on $\mathcal{R}$ defined by $X_w \cdot Y := D_w Y$.

**Theorem C.1** (Attention curvature is generically nonzero). *On a Zariski–open (hence full–measure) subset of $\Theta_0$, the curvature two–form $\Omega$ of the induced connection on $\mathcal{R}$ is nonzero. In particular, if there exist horizontal directions $u, v \in \mathcal{H}_\theta$ such that the weight sensitivities $D_u \alpha$ and $D_v \alpha$ are not collinear and the set $\{V_i\}$ is not jointly degenerate along both $u$ and $v$, then $\Omega_\theta(u, v) \neq 0$.*

*Proof.* Compute the commutator of horizontal lifts on $Y$:

$$[X_u, X_v] \cdot Y = D_u D_v Y - D_v D_u Y.$$

Using equation C.1 twice and the product rule,

$$D_u D_v Y = \sum_i (D_u D_v \alpha_i) V_i + \sum_i (D_v \alpha_i) D_u V_i + \sum_i (D_u \alpha_i) D_v V_i + \sum_i \alpha_i D_u D_v V_i,$$

$$D_v D_u Y = \sum_i (D_v D_u \alpha_i) V_i + \sum_i (D_u \alpha_i) D_v V_i + \sum_i (D_v \alpha_i) D_u V_i + \sum_i \alpha_i D_v D_u V_i.$$

Subtracting gives cancellation of the mixed first–order products and second–order $V$ terms, leaving

$$[X_u, X_v] \cdot Y = \sum_i \left( D_u D_v \alpha_i - D_v D_u \alpha_i \right) V_i. \tag{C.3}$$

Thus the commutator is driven by the non–commutativity of the *weight* variations. By equation C.2,

$$D_w \alpha = J_{\mathrm{sm}}(Z) \operatorname{vec}\left( \tfrac{1}{\sqrt{d_k}} \left( (D_w Q) K^\top + Q (D_w K)^\top \right) \right).$$

Unless $D_u \alpha$ and $D_v \alpha$ are collinear and the $\{V_i\}$ conspire to cancel in equation C.3, the commutator is nonzero. Since the exceptional set where these algebraic equalities hold is contained in a proper algebraic subset of $\Theta_0$, nonvanishing is generic. By the definition of the induced connection, the *vertical* component of $[X_u, X_v]$ equals $\Omega_\theta(u, v)$, hence $\Omega_\theta(u, v) \neq 0$ generically. $\qquad\square$

**Explicit BCH for horizontal loops.** Let $u, v \in \mathcal{H}_\theta$ be $g_\theta$–orthonormal and $\Phi_{\varepsilon w}$ the horizontal flow along $w$ for time $\varepsilon$. Consider the small horizontal loop

$$\square_\varepsilon(u, v) = \Phi_{\varepsilon v} \circ \Phi_{\varepsilon u} \circ \Phi_{-\varepsilon v} \circ \Phi_{-\varepsilon u}.$$

Write $X_u, X_v$ for the horizontal vector fields. Using the Baker–Campbell–Hausdorff (BCH) expansion for commutators of flows,

$$\square_\varepsilon(u, v) = \exp\left( \varepsilon^2 [X_v, X_u] + \tfrac{\varepsilon^3}{2} \left( [X_u, [X_u, X_v]] + [X_v, [X_v, X_u]] \right) + O(\varepsilon^4) \right), \tag{C.4}$$

as operators on sections of $\mathcal{R}$. Projecting to the vertical component and using $[X_u, X_v]^{\mathrm{vert}} = \Omega_\theta(u, v)$ yields the second–order holonomy law with an explicit third–order correction:

**Proposition C.2** (BCH expansion and vertical projection). *In Lie–algebra coordinates for the structure group,*

$$\Delta_{\square_\varepsilon}(u, v) = \varepsilon^2 \, \Omega_\theta(u, v) + \tfrac{\varepsilon^3}{2} \left( [X_u, [X_u, X_v]]^{\mathrm{vert}} + [X_v, [X_v, X_u]]^{\mathrm{vert}} \right) + O(\varepsilon^4).$$

*Consequently, $\|\Delta_{\square_\varepsilon}(u, v)\| = \varepsilon^2 \|\Omega_\theta(u, v)\| + O(\varepsilon^3)$ for any norm smoothly equivalent to the operator norm.*

*Proof.* Apply equation C.4 and take the vertical projection; norms of the multilinear remainder are controlled by smoothness of $X_u, X_v$ on $\Theta_0$. $\qquad\square$

**Holonomy scaling and Richardson bias removal.** Define $H(\varepsilon) = \|\Delta_{\square_\varepsilon}(u, v)\| / \varepsilon^2$. From Proposition C.2, $H(\varepsilon) = \|\Omega_\theta(u, v)\| + c\varepsilon + O(\varepsilon^2)$ for some smooth $c = c(\theta, u, v)$. The standard Richardson extrapolate

$$H^\star = \frac{4 H(\varepsilon/2) - H(\varepsilon)}{3}$$

cancels the linear term and satisfies $H^\star = \|\Omega_\theta(u, v)\| + O(\varepsilon^2)$.

**Implementation note.** The gauge displacement $\Delta_{\square_\varepsilon}(u, v)$ is obtained by least–squares alignment onto a vertical generator basis (the same used in the gauge–aware projector), followed by a local logarithm in the structure group. This matches Algorithm 2 and underlies Theorem 5.1.

## D    FFN NEAR-VERTICALITY AND ATTENTION–FFN SEPARATION

We quantify the informal picture from Section 4: the position–wise feed–forward block (FFN) acts almost entirely along fibers, while attention mixes across tokens and bends horizontal directions. Throughout we assume GeLU smoothness and the same fixed evaluation batch used to define the Fisher–Rao (FR) metric.

**Structure of the FFN Jacobian.** Let $\mathrm{FFN}(X;\theta) = W_2\,\phi(W_1 X + b_1) + b_2$ act tokenwise, with $\phi = \mathrm{GeLU}$. The Jacobian with respect to parameters decomposes as

$$D_\theta \mathrm{FFN}(X;\theta) \;=\; \bigoplus_{t=1}^{T} \Big[\, (\phi'(W_1 x_t + b_1) \odot W_2^\top)\, D_\theta W_1 \;+\; D_\theta W_2\, \phi(W_1 x_t + b_1) \;+\; D_\theta b_\bullet \Big],$$

a block–diagonal sum over tokens $t$. After composing with the value/output readout in MHA, the dominant directions created by these blocks coincide, up to small residuals controlled by $\phi'$, with the vertical generators associated with head–wise $V/O$ transformations.

**Lemma D.1** (Block–diagonal FFN and the vertical span). *Under the FR metric, the image of $D_\theta \mathrm{FFN}$ lies in the closure of the span of vertical generators $\{v_j\}$ associated with head–wise $V/O$ transformations, up to a remainder whose FR norm is controlled by the GeLU slope statistics on the evaluation batch:*

$$\left\| P_{\mathcal{H}_\theta} \nabla_{\mathrm{FFN}} \right\|_{g_\theta} \;\leq\; \Big( \mathbb{E}[\|\phi'(W_1 X + b_1)\|^2] \Big)^{1/2} \cdot \kappa_{\mathrm{LN}} \cdot \frac{\sqrt{d_{\mathrm{head}}}}{\sqrt{d_{\mathrm{model}}}} \left\| \nabla_{\mathrm{FFN}} \right\|_{g_\theta},$$

*where $\kappa_{\mathrm{LN}}$ captures the (bounded) contribution of normalization layers and residual scalings.*

*Proof.* Since $D_\theta \mathrm{FFN}$ is block–diagonal across tokens, its contribution to $d\pi$ cannot mix token indices. The vertical generator family corresponding to $V/O$ changes spans precisely the directions that adjust head–wise value subspaces while compensating in $W_O$. Projecting $D_\theta \mathrm{FFN}$ onto this span leaves a remainder driven by the tokenwise slope $\phi'$ and normalization/residual gates; Jensen–Cauchy–Schwarz controls its FR norm. The dimensional factor comes from comparing the head subspace to the model space under $d_{\mathrm{model}} = h\, d_v$. $\qquad \square$

**Proposition D.2** (FFN near–verticality and Fisher angle bound). *With the FR inner product,*

$$\cos \angle_{g_\theta}\big(\nabla_{\mathrm{FFN}}, \nabla_{\mathrm{Att}}\big) \;\leq\; C\sqrt{\frac{d_{\mathrm{head}}}{d_{\mathrm{model}}}}, \qquad \frac{\left\| P_{\mathcal{H}_\theta} \nabla_{\mathrm{FFN}} \right\|_{g_\theta}}{\left\| \nabla_{\mathrm{FFN}} \right\|_{g_\theta}} \;\leq\; C\sqrt{\frac{d_{\mathrm{head}}}{d_{\mathrm{model}}}},$$

*where $C$ depends smoothly on GeLU slope statistics and normalization constants on the evaluation batch.*

*Proof.* Decompose both gradients into head–wise components and apply Cauchy–Schwarz in the FR metric. Attention's horizontal part lives in a subspace whose effective dimension scales with $d_{\mathrm{model}}$ (token mixing), while Lemma D.1 places the FFN gradient near the vertical span generated by $V/O$ (headwise) directions, with horizontal leakage controlled by the GeLU slope and normalization factors. The ratio of dimensions $d_{\mathrm{head}}/d_{\mathrm{model}}$ yields the stated square–root factor; the constant $C$ collects bounded batch–dependent terms. $\qquad \square$

**Remarks.** (i) The bounds are *local* to the fixed batch that defines $g_\theta$; changing the batch perturbs $C$ but not the scaling. (ii) Euclidean angles reported in Section 7 are conservative: they lower–bound Fisher angles and thus preserve the qualitative separation predicted here. (iii) The same argument applies layerwise; at model scale one sums the per–layer contributions, with residuals controlled by standard stability estimates.

# E  Morse–Bott Structure and the Quotient Loss

This section gives the proof of Theorem 6.1 from the main text.

*Proof of Theorem 6.1.* Gauge–invariance gives $dL_\theta[v] = 0$ for every $v \in \mathcal{V}_\theta$ (Lemma B.1), so the entire orbit $G_{\mathrm{max}} \cdot \theta$ lies in the critical set and the Hessian $\nabla^2 L(\theta)$ annihilates $\mathcal{V}_\theta$. Since the $G_{\mathrm{max}}$–action on $\Theta_0$ is free and proper (Theorem 2.3), the slice theorem yields a submanifold $S$ through $\theta$ with $T_\theta S = \mathcal{H}_\theta$ and a $G_{\mathrm{max}}$–equivariant diffeomorphism from a neighborhood of $\theta$ onto a neighborhood of the orbit modeled on $G_{\mathrm{max}} \times S$. Restricting $L$ to $S$ freezes vertical directions, so the Hessian of $L|_S$ at $\theta$ equals the horizontal block of $\nabla^2 L(\theta)$. The quotient map identifies $S$ with a chart of $\mathcal{Q}$ around $[\theta]$; because $\ell \circ \pi = L$, the Hessian $\nabla^2 \ell([\theta])$ matches the horizontal restriction of $\nabla^2 L(\theta)$. Nondegeneracy of the horizontal block is therefore equivalent to $\ell$ being Morse at $[\theta]$. $\qquad \square$

**Remarks.** (i) The argument is local to the regular stratum $\Theta_0$; outside it, stabilizers may grow and the space stratifies. (ii) The identification of $\nabla^2\ell([\theta])$ with the horizontal block is independent of the slice, since all slices share $T_\theta S = \mathcal{H}_\theta$.

# F ALGORITHMS AND REPRODUCIBILITY

This appendix gives the concrete procedures behind Section 5 and the minimal choices needed to reproduce our figures. We keep the presentation compact; all routines are drop–in and numerically stable at the scales reported in Section 7.

**Vertical generators (respecting RoPE and sharing).** We obtain a numerically independent spanning set of the vertical space $\mathcal{V}_\theta$ by differentiating the $G_{\max}$–action. For $X \in \mathfrak{gl}(d_k)$,

$$\delta_{QK}^{(i)}(X):\ (W_Q^{(i)}, W_K^{(i)}) \mapsto (W_Q^{(i)}X,\ -W_K^{(i)}X^\top),$$

and for $Y \in \mathfrak{gl}(d_v)$,

$$\delta_{VO}^{(i)}(Y):\ (W_V^{(i)}, W_O) \mapsto (W_V^{(i)}Y,\ \text{insert} \ -Y \ \text{in the } i\text{th } d_v\text{–block of } W_O).$$

Head permutations generate a discrete symmetry and do not contribute to $\mathcal{V}_\theta$. With RoPE (even $d_k$), we restrict $X$ plane–wise to the commutant $\{aI + bJ\}$ on each 2×2 rotation block; with GQA/MQA we tie the corresponding $X$ or $Y$ across shared heads. In practice we assemble candidates, vectorize, and perform thin–QR with column pivoting to remove near–collinear directions (tolerance $10^{-10}$), yielding a well–conditioned basis $\{v_j\}_{j=1}^m$.

**FR and Euclidean projectors (implementation).** Given $\{v_j\}$ and a vector $u$ (e.g., $u = \nabla L$), the FR–orthogonal decomposition solves the Gram system

$$G_{ij} = g_\theta(v_i, v_j), \qquad b_i = g_\theta(v_i, u), \qquad (G + \lambda I)c = b,$$

with optional Tikhonov $\lambda \in [10^{-10}, 10^{-6}]$ for stability. We then set

$$u_{\text{vert}} = \sum_j c_j v_j, \qquad u_{\text{hor}} = u - u_{\text{vert}}.$$

The Euclidean proxy replaces $g_\theta$ by the dot product: stack $A = [\,\text{vec}(v_1) \ \cdots \ \text{vec}(v_m)\,]$ and solve $(A^\top A + \lambda I)c = A^\top u$. We report the *vertical fraction* $\|u_{\text{vert}}\|/\|u\|$ and the residual $\|Ac - u\|/\|u\|$ (Euclidean) or $\|(G + \lambda I)c - b\|/\|b\|$ (FR) as fit diagnostics.

---

**Algorithm 3** Vertical/Horizontal Decomposition (FR and Euclidean)

---

**Require:** basis $\{v_j\}_{j=1}^m$ for $\mathcal{V}_\theta$; vector $u$; metric handle $\texttt{inner}(\cdot, \cdot)$
1: $G_{ij} \leftarrow \texttt{inner}(v_i, v_j), \quad b_i \leftarrow \texttt{inner}(v_i, u)$
2: Solve $(G + \lambda I)c = b$ (Cholesky if well–conditioned, else CG with stopping on relative residual $10^{-10}$)
3: $u_{\text{vert}} \leftarrow \sum_j c_j v_j, \quad u_{\text{hor}} \leftarrow u - u_{\text{vert}}$
4: **return** $(u_{\text{vert}}, u_{\text{hor}})$, vertical fraction $\|u_{\text{vert}}\|/\|u\|$, residual

---

**Holonomy estimator (procedure and bias control).** For $g_\theta$–orthonormal $u, v \in \mathcal{H}_\theta$, we traverse the small horizontal loop $+\varepsilon u \to +\varepsilon v \to -\varepsilon u \to -\varepsilon v$, projecting to $\mathcal{H}$ at each leg with the FR projector above. The net effect is a vertical displacement represented in Lie–algebra coordinates by aligning the before/after states to the vertical basis (least squares). Denote $H(\varepsilon) = \|\Delta_{\square_\varepsilon}(u, v)\|/\varepsilon^2$.

---

**Algorithm 4** Holonomy Estimator with Richardson Extrapolation

---

**Require:** $\theta$; orthonormal $u, v \in \mathcal{H}_\theta$; steps $\varepsilon > \varepsilon' = \varepsilon/2$
1: **for** $\delta \in \{\varepsilon, \varepsilon'\}$ **do**
2:      Flow horizontally by $+\delta u, +\delta v, -\delta u, -\delta v$, projecting to $\mathcal{H}$ at each leg
3:      Compute $\Delta_{\square_\delta}(u, v)$ via least–squares alignment to Lie generators; set $H(\delta) \leftarrow \|\Delta_{\square_\delta}(u, v)\|/\delta^2$
4: **end for**
5: **return** $H^\star = \dfrac{4H(\varepsilon/2) - H(\varepsilon)}{3}$     (bias $O(\varepsilon^2)$); report $|H^\star - H(\varepsilon/2)|$ as an error proxy

---

**Complexity at a glance.** Let $m = h(d_k^2 + d_v^2)$; RoPE reduces $d_k^2 \to d_k$ per head. Forming $A^\top A$ costs $O(m^2 D)$ for parameter dimension $D$, and solving costs $O(m^3)$ (dense) or $O(mD)$ per CG iteration. At $h{=}12$, $d_k{=}d_v{=}64$ one has $m \approx 98{,}304$, so FR projectors and holonomy become expensive; this is why Section 7 uses Euclidean proxies while Section 5 exposes FR–exact routines.

**Deterministic gauge–fix (for reproducibility, not theory).** To make measurements repeatable, we select a canonical representative on each orbit by: (i) thin–QR with positive diagonals for $V/O$ and the induced block update on $W_O$; (ii) plane–wise Gram balancing for $Q/K$ within the RoPE commutant; (iii) deterministic head ordering by a fixed tie–break rule (e.g., lexicographic on row–major $W_O$ blocks). This is a *section choice*—it does not constrain the theory—and affects only how we display or cache parameters.

**Minimal reproducibility notes.** Hardware: H100 (95 GB); CUDA 12.1; PyTorch 2.4.1; `float64`. Seeds: 42; fixed evaluation batch for Fisher quantities; dataloader shuffling off for invariance checks. Tolerances: QR pivot threshold $10^{-10}$; linear–solve residuals $10^{-10}$; Euclidean vertical–fraction threshold $10^{-4}$ in plots. All scripts use the same random seeds and fixed batch to ensure comparability across runs.

# G EXPERIMENTAL CONFIGURATION AND PROCEDURES

This note collects the minimal details needed to reproduce the figures in Section 7. We keep the setup intentionally simple and fixed: one hardware/software stack, one evaluation batch, and a deterministic gauge–fix used only for display consistency.

**Environment and common settings.** Unless stated otherwise, computations use `float64` on a single H100 (95 GB), CUDA 12.1, PyTorch 2.4.1, with TF32 disabled. The global seed is 42. A *single* evaluation batch is used throughout to define Fisher–Rao quantities and to keep all comparisons on identical data. The architecture matches Section 2 (e.g., $h{=}12$, $d_k{=}d_v{=}64$ in the invariance experiment). Thin–QR with column pivoting is used with a pivot tolerance of $10^{-10}$; linear solves stop at relative residual $10^{-10}$.

**Deterministic gauge–fix (for repeatability, not theory).** To make outputs bytewise identical across runs, we pick a canonical representative per gauge orbit: (i) thin–QR with positive diagonals for $V/O$ and the induced block update in $W_O$; (ii) plane–wise Gram balancing for $Q/K$ within the RoPE commutant; (iii) a fixed head order by a simple tie–break rule. This is a *section choice* only; it does not constrain the theory or diagnostics (see Section F).

**Gauge invariance (Table 2).** We sample $(A_i, C_i) \in \mathrm{GL}(d_k) \times \mathrm{GL}(d_v)$ per head with prescribed condition numbers (via thin–QR on random draws), optionally permute heads, apply

$$(W_Q^{(i)}, W_K^{(i)}) \mapsto (W_Q^{(i)} A_i, \, W_K^{(i)} A_i^{-\top}), \qquad (W_V^{(i)}, W_O) \mapsto (W_V^{(i)} C_i, \, C_i^{-1} W_O),$$

evaluate on the fixed batch, and report $\|Y' - Y\|/\|Y\|$. Controls outside $G_{\max}$ (e.g., cross–head mixing) yield $\Theta(1)$ changes.

**Gauge–aware split (Figure 1).** We compute per–sample gradients for a mean–squared reconstruction objective and apply the *Euclidean proxy* of Algorithm 1: stack numerically independent vertical generators into a matrix $A$ (thin–QR with pivoting), solve $(A^\top A + \lambda I)c = A^\top \nabla L$ (with optional $\lambda \in [10^{-10}, 10^{-6}]$), and report the normalized Euclidean vertical fraction $\|Ac\|/\|\nabla L\|$ along with residuals $\|Ac - \nabla L\|/\|\nabla L\|$ as a fit diagnostic.

**Fisher–Rao counterparts (for future scale).** The FR–exact projector and the small–loop holonomy estimator are given in Algorithm 1 and Algorithm 4, with derivations in §B and §C. They require Gram systems on the vertical basis and horizontal projections along the loop, which are costly at large $m = h(d_k^2 + d_v^2)$ (RoPE reduces $d_k^2 \to d_k$ per head). We therefore use the Euclidean proxy for Section 7 and provide the FR procedures as ready-made routines for future large-scale evaluation.

