# OpenReview forum: "Gauge Fiber Bundle Geometry of Transformers"
_ICLR.cc/2026/Conference — ICLR 2026 Conference Withdrawn Submission_

### Official Review · Reviewer_iiue · 2025-10-28

**Soundness:** 2
**Presentation:** 1
**Contribution:** 1
**Rating:** 2
**Confidence:** 3

**Summary:**

This paper aims to use a gauge theory framework to better understand transformers. It does this by interpreting the weight to function map within a principal bundle framework. In particular, it decomposes the tangent space (with respect to Fisher–Rao) around a generic point in the weight space $\Theta_0$ into directions that change the realized function and directions that are simply parameter symmetries that don’t change the function. The idea is that understanding this distinction helps us better understand the loss landscape and ultimately the behavior of transformers. To explore this, the paper first presents some results around weight symmetries in attention heads. Then it performs some calculations of curvature for feed forward and attention blocks. Finally, the paper introduces several computational approaches to approximating the constructions it has introduced and runs small-scale experiments.

**Strengths:**

**A refreshing injection of ideas that have been powerful in physics:** Fiber bundles, gauge symmetries, holonomy, and other constructions from differential geometry have proven to be immensely valuable in physics. It is an interesting research direction to explore how these can be used in machine learning, especially as machine learning systems scale and new abstractions are needed.

**The premise of decomposing $T\Theta_0$ is intriguing:** The direction taken by this paper, which focuses on the geometry of the loss landscape, seems reasonable and the reviewer is interested in seeing this further developed. There have been several approaches that have focused on optimization and parameter symmetries and it would be useful to understand how this paper relates to those [1].

[1] Zhao, Bo, et al. "Symmetry teleportation for accelerated optimization." Advances in neural information processing systems 35 (2022): 16679-16690.

**Weaknesses:**

**Who is the audience for this paper?:** It is this reviewer’s impression that this paper is written for an audience of geometry or physics experts. In the introduction, rather than explaining what gauge theory is (at a high-level for those that haven’t heard of it), the motivation for applying it to transformers, and what machine learning-relevant questions the paper aims to answer, the paper simply launches into formalism, expecting the reader to be comfortable with ‘gauge orbits’, ‘principle connections’, and ‘holonomy’. To this reviewer’s knowledge, this will exclude the vast majority of the ICLR community from even understanding at a high level what the paper is about. While technical depth and sophistication should not be a barrier to acceptance at ICLR, some effort should be made to at least make the point of the paper intelligible to those outside of a narrow subdomain. The introduction is a good place to do this. Then, the body of the paper can dive into much greater depth while asking more of the reader.

**Answering the ‘so-what’ for machine learning:** Very little space is spent discussing how the results of the paper shed light on problems in machine learning beyond being able to decompose $T\Theta_0$. Use of complicated theoretical machinery should be justified by the fact that it enables researchers to access information that would be hard to obtain via simpler tools. For example, what does the notion of holonomy allow us to say about transformers that we could not say before. For the case of curvature, a cryptic remark is made about it measuring ‘context sensitivity’ but this is otherwise left unexplained.

**A lot of notation is not defined:** To take as an example the Introduction:
- Line 036: $\mathcal{Q}$ is not defined. It is a function space presumably, but which one? What is its topology?
- Most of the operators on Line 049 are left undefined. (Presumably $L$ is the loss but $L$ is also used for the depth in Line 079).
- Line 094: The paper should state what $g$ is. E.g., $g \in G$.
- The property of being *Morse* is central to Theorem 6.2, but it is never defined.

**Corollary 2.6:** Generally, it is not the case that the parameter symmetry group for a model as a whole is equal to the product of symmetry groups of each layer (or block). See for example symmetries that come from the computation graph [2] (to this reviewer’s knowledge, this particular question has not been explored for the transformer architecture). As such, this reviewer believes that Corollary 2.6 needs a proof.

**Nitpicks**
- Line 194: ‘FFN’ is only defined in the appendix

[2] Lim, Derek, et al. "Graph metanetworks for processing diverse neural architectures." arXiv preprint arXiv:2312.04501 (2023).

**Questions:**

- Why is the acronym for *feed forward block* FFN?
- Line 284: “path dependence predicted by curvature”. What is the meaning of path dependence in this setting? What is the reason one would care about it? Earlier it was said that this ‘path dependence’ captures the impact on context, can this be expanded on?
- How would the proposed diagnostics be used in practice?

---

### Official Review · Reviewer_19Br · 2025-10-31

**Soundness:** 4
**Presentation:** 2
**Contribution:** 2
**Rating:** 4
**Confidence:** 2

**Summary:**

Caveat: I am not an expert in this area, and have tried my best to understand the results in this paper.

This paper seeks to exactly parametrize the gauge symmetries in a multi-head attention layer, which are the sets of layer parameters that map to the same function represented by the layer. The main result is the quantification of the maximal gauge group (i.e. the largest set of parameter symmetries). Once these symmetries have been identified, the authors define a partition of the tangent space into a (vertical) subspace tangent to the gauge orbit (ie, where the function remains constant) and a (horizontal) orthogonal one (where orthogonality is defined by the Fisher-Rao metric). This allows defining the natural gradient by simply projecting the parameter gradient to the horizontal subspace. The authors construct an algorithm for performing this decomposition up to arbitrary error. Next, they define an estimator for the holonomy of the connection defined by attention (which is essentially a way to measure how curved the parameter space is). With empirical evidence using the Euclidean metric, they show that their gauge symmetries are indeed preserved, and the natural gradient indeed lies along the horizontal (function-changing) subspace.

**Strengths:**

The paper formalizes a framework to think about parameter symmetries and natural gradient descent in neural networks in a general way, which will be useful for other architectures. I enjoyed the functional/coordinate-free approach of this paper. The paper is also precise in its scope of the object it is studying.

**Weaknesses:**

The paper in its current form is extremely terse. I believe a lot of technical machinery is introduced in this paper but it is not clear why such machinery is essential (eg. for proving Theorem 2.1). The amount of technical detail makes it not super clear what the main results in this paper are; is it simply the characterization of the maximal gauge group for multi-head attention, and the construction of the holonomy estimator? The authors should try to motivate the significance of their results more. For example, are there limitations in current optimizers for multi-head attention? Can we possibly construct better optimizers for transformers given their results? This would help situate the results of this paper in the broader literature.

**Questions:**

* One piece of terminology confuses me: why do you refer to $\mathcal{V}$ as the vertical component, and $\mathcal{H}$ as the horizontal component? The picture I have is that the gauge orbit represents the surface on which the network function doesn't change; but $\mathcal{H}$ is normal to this surface and hence 'vertical' to it, making $\mathcal{V}$ the horizontal component. This is just a matter of terminology but I was confused by it.


* Several quantities eg. $d_v, d_k, h$ in Line 45 are only defined later (eg. in Definition 2.1). Several quantities (such as induced gauge displacement eg. $∆_{\epsilon}(u, v)$) should be defined.

---

### Official Review · Reviewer_dLCE · 2025-10-31

**Soundness:** 3
**Presentation:** 4
**Contribution:** 3
**Rating:** 4
**Confidence:** 3

**Summary:**

This paper offers a novel, geometry-first perspective on Transformer models utilizing the GeLU activation function. The authors describe the Transformer's parameter space as a principal gauge fiber bundle. The parameter space is decomposed into Fibers and Horizontals. The fibers are composed of gauge orbits of the head-wise symmetry group, representing the set of functionally equivalent models. The horizontals represent the training directions in the parameter space that actually change the model's function. An empirical Fisher information metric (Fisher-Rao metric) is introduced, providing a canonical horizontal distribution for the parameter space.

**Strengths:**

This work provides a profound and clear mathematical foundation at a theoretical level for the structure and training dynamics of Transformer models. It aids in the understanding of model redundancy (gauge symmetry) and how to geometrically optimize training paths.

**Weaknesses:**

Many advanced concepts are introduced, making the paper technically dense to general readers, although it is well-written.
This paper lacks empirical validation and/or practical impact.

**Questions:**

Why is GeLU important? Can this great framework be generalized to other activations?

How strong are the assumptions, e.g., Def. 2.1, in the real world?

---

### Note · Authors · 2025-11-12

**Comment:**

I literally just learned that a version of this paper was accepted by NeurRep workshop on 12/7/25 and will be part of the proceedings. So I will publish this work at that venue.

Thanks for the great feedback from the reviewers

**Withdrawal Confirmation:**

I have read and agree with the venue's withdrawal policy on behalf of myself and my co-authors.